# Optimization of a WGA-Free Molecular Tagging-Based NGS Protocol for CTCs Mutational Profiling

**DOI:** 10.3390/ijms21124364

**Published:** 2020-06-19

**Authors:** Giuseppa De Luca, Barbara Cardinali, Lucia Del Mastro, Sonia Lastraioli, Franca Carli, Manlio Ferrarini, George A. Calin, Anna Garuti, Carlotta Mazzitelli, Simona Zupo, Mariella Dono

**Affiliations:** 1Molecular Diagnostic Unit, IRCCS Ospedale Policlinico San Martino, 16132 Genova, Italy; giuseppa.deluca@hsanmartino.it (G.D.L.); sonia.lastraioli@hsanmartino.it (S.L.); simonettazu@hotmail.it (S.Z.); 2Breast Unit, IRCCS Ospedale Policlinico San Martino, 16132 Genova, Italy; barbara.cardinali@hsanmartino.it (B.C.); lucia.delmastro@hsanmartino.it (L.D.M.); carlotta.mazzitelli@edu.unige.it (C.M.); 3Department of Internal Medicine (Di.M.I.), University of Genova, 16132 Genova, Italy; anna.garuti@unige.it; 4Department of Pathology, IRCCS Ospedale Policlinico San Martino, 16132 Genova, Italy; franca.carli@hsanmartino.it; 5DIMES, Anatomy Section, Medical School, University of Genova, 16132 Genova, Italy; ferrarini.manlio@gmail.com; 6Department of Experimental Therapeutics, The University of Texas MD Anderson Cancer Center, Houston, Texas, TX 77030, USA; gcalin@mdanderson.org

**Keywords:** Circulating Tumor Cells, breast cancer, liquid biopsy, Next-Generation Sequencing, molecular tagging, Whole Genome Amplification-free, single-cell genomics, biomarkers, precision medicine

## Abstract

Molecular characterization of Circulating Tumor Cells (CTCs) is still challenging, despite attempts to minimize the drawbacks of Whole Genome Amplification (WGA). In this paper, we propose a Next-Generation Sequencing (NGS) optimized protocol based on molecular tagging technology, in order to detect CTCs mutations while skipping the WGA step. MDA-MB-231 and MCF-7 cell lines, as well as leukocytes, were sorted into pools (2–5 cells) using a DEPArray™ system and were employed to set up the overall NGS procedure. A substantial reduction of reagent volume for the preparation of libraries was performed, in order to fit the limited DNA templates directly derived from cell lysates. Known variants in *TP53*, *KRAS*, and *PIK3CA* genes were detected in almost all the cell line pools (35/37 pools, 94.6%). No additional alterations, other than those which were expected, were found in all tested pools and no mutations were detected in leukocytes. The translational value of the optimized NGS workflow is confirmed by sequencing CTCs pools isolated from eight breast cancer patients and through the successful detection of variants. In conclusion, this study shows that the proposed NGS molecular tagging approach is technically feasible and, compared to traditional NGS approaches, has the advantage of filtering out the artifacts generated during library amplification, allowing for the reliable detection of mutations and, thus, making it highly promising for clinical use.

## 1. Introduction

Circulating Tumor Cells (CTCs) in the bloodstream, which originate from primary and/or metastatic tumors, are both phenotypically and molecularly highly heterogeneous [1,2].

An increasing number of data have confirmed the role of CTCs count and their molecular characterization in patient prognosis, relapse prediction, and therapy outcome, indicating that CTCs represent an attractive blood-based biomarker with high potential for understanding the clinical behavior of the associated disease [3,4,5,6,7,8,9,10].

To date, major efforts towards the molecular characterization of CTCs have been focused on the optimization of genetic analysis protocols, mainly including Whole Genome Amplification (WGA) as a first step, which allows for the isolation of an acceptable amount of DNA from few starting molecules [11,12,13,14,15,16]. Unfortunately, different studies have demonstrated a series of drawbacks arising from WGA procedures, such as allelic dropout, chimeric molecules, low uniformity, and polymerase errors, which lead to bias in the interpretation of molecular data [17,18,19,20,21]. These biases are partly caused by an unavoidable delay in the mutational profile analysis of CTCs in routine laboratory practice. 

Meanwhile, different workflows have been developed to provide alternative options to the traditional techniques which do not require this preamplification step, instead of utilizing downstream whole-genome or targeted sequencing [22,23,24].

This work proposes a modified workflow for a targeted Next-Generation Sequencing (NGS) panel, which already was tested in our laboratory for the molecular characterization of Circulating Tumor DNA (ctDNA) and which is based on molecular tagging technology. This approach is capable of intercepting even single DNA molecules in difficult biological samples, such as ctDNA. The original protocol was adapted for NGS analysis of few pooled cells isolated on a DEPArray^TM^ system, with the overall aim of completely skipping WGA; instead, directly proceeding with library construction while obtaining reliable sequencing results.

## 2. Results

### 2.1. DEPArray^TM^ Sorting and Validation of Optimized NGS Procedures

#### 2.1.1. Cell Detection and DNA Integrity Assessment

MDA-MB-231 (MDA) and MCF-7 cell lines were employed as surrogate Breast Cancer (BC) CTCs and used in spiking-in experiments by addition to a healthy donor blood sample before DEPArray^TM^ isolation. 

Sorting was realized on the basis of morphological and phenotypic features of MDA/MCF-7 cells: they presented an oval shape and round features and high nuclear–cytoplasmic ratio. Furthermore, they variably expressed Epithelial Cell Adhesion Molecule (EpCAM) and Cluster of Differentiation 44 (CD44) markers, the expression of the latter being almost complete (95–100%) in MDA, but almost negative in MCF-7 (~1–2%); whereas EpCAM was, on average, expressed by 45% of MDA and on the totality of MCF-7 cultured cells, respectively. Neither of the two cell lines showed a Cluster of Differentiation 45 (CD45) expression (by Fluorescent-Activated Cell Sorter, FACS, data not shown). Leukocytes were mostly identified and isolated on the basis of their small size and high CD45 expression. Examples of DEPArray^TM^ sorted and identified cells are reported in Figure 1.

To check for DNA integrity, MDA and leukocytes were subsequently isolated on DEPArray^TM^ and a total of 18 samples deriving from two independent experiments were analyzed (i.e., nine MDA cells and nine leukocytes).

Then, DNA lysates from cells pools composed of one, two, and five cells (each run tested in triplicate) were checked using Polymerase Chain Reaction (PCR) by amplification of four chromosomal regions of different lengths (91, 108–166, 299, and 614 bp, respectively of 12 p, 5 p, 17 p, and 6 p chromosomes), where PCR products were visualized by electrophoresis. Optimal DNA integrity was demonstrated by the presence of four bands, respectively, in 8/9 MDA (88.9%) and 7/9 leukocytes (77.8%) samples (Figure 2). 

However, as downstream NGS analysis is ensured only for cells with at least three genomic amplicons (as per Ampli1 QC kit manufacturer instructions and [25]), our preparation and cell isolation procedures demonstrated outperformance, as 83% (15/18) of tested samples were considered potentially suitable for successive NGS experiments. 

The remaining three samples—composed of one cell (one sample) or two cells (two samples) and derived from MDA cells in one case and from leukocytes in the other two—displayed only two amplified bands, thus showing that loss of isolated cell/s during experimental manipulation did not occur, although some genetic loss/fragmentation could be detected. Consequently, these data confirmed the robustness and reliability of the cell recovery step, as well as that of the whole sorting procedure.

#### 2.1.2. From Canonical to Optimized Molecular Tagging NGS Workflow: Experimental Setting and Development

In the preliminary NGS assessment, MDA lysates of one, three, and five cells were added to 5 ng of normal DNA. Then, these DNA mixtures were employed to prepare libraries using the Oncomine Breast cfDNA Research Assay v2 (OBcfRAv2) NGS targeted panel. Two hotspots present in the MDA cell line—the *KRAS* p.Gly13Asp and *TP53* p.Arg280Lys—were followed and detected.

In particular, the library prepared starting from three MDA cells showed 0.61% and 1.06% molecular allele frequencies (MAFs) for *KRAS* p.Gly13Asp and *TP53* p.Arg280Lys, respectively; likewise, libraries prepared from five MDA cells displayed 0.67% and 0.95% MAFs for the two variants, respectively. Interestingly, variants detection was still feasible from one MDA cell lysate, despite the fact that the addition of 5 ng of normal genomic DNA resulted in 1000× dilution of the ~6 pg mutated alleles, where MAFs obtained for *KRAS* p.Gly13Asp and *TP53* p.Arg280Lys were 0.18% and 0.38%, respectively (data not shown).

Collectively, these preliminary experiments were encouraging, as they demonstrated that tagging NGS is a robust and highly sensitive approach for mutation detection at the single-cell level, being able to discriminate one mutated allele within a bulk of wild-type alleles. 

Consequently, we optimized the sequencing protocol by directly using isolated MDA cells as a unique starting genetic material, in the complete absence of normal genomic DNA or WGA, for subsequent library preparation. 

As this strategy can dramatically decrease the input amount of DNA, an adjustment of the NGS protocol was developed. The entire new procedure presented main technical changes consisting of the following four items: (1) DNA derived only from sorted cells was used as a starting material for the preparation of libraries; (2) a three-times volume reduction of the reaction reagents was applied for the two PCRs planned by the OBcfRAv2 protocol, as reported in detail in Appendix A; (3) libraries were mixed up to 24 samples for Ion 520™ Chip loading (low-coverage sequencing, Table 1). 

The modified protocol was tested not only on MDA cell lines, but also on pools of MCF-7 cells sorted and collected as already described, in order to verify the protocol feasibility in a second cell line harboring different mutations.

By applying the optimized NGS protocol, a total of 47 libraries were prepared from nine different cellular pool types. In particular, three pools composed of two, four, and five tumor cells only and four combined pools (4:1, 3:2, 2:3, and 1:4 MDA/leukocytes ratios, respectively) were investigated for MDA, together with two pools of leukocytes only as negative controls. For MCF-7 analysis, duplicates were considered for all pools, except for pools consisting of two cells only and the 1:4 MCF-7/leukocytes combination, respectively, which were tested in triplicate. In this context, the cellular combinations at different ratios were intended to test the detection accuracy of mutated alleles in the presence of wild-type ones.

First, sufficient amounts for subsequent massive sequencing (range 11–246 pMol, median 65 pMol) were obtained in 45/47 samples (95.7%) and only two libraries failed to be quantified (from MDA cells/leukocytes combinations at 3:2 and 2:3 ratios), probably indicating a complete loss of genetic material in these samples. Subsequently, we decided to analyze two new MDA cellular preparations to complete the triplicates and, thus, a total of 49 libraries were prepared, with a final performance of 47/49 samples (95.9%).

Second, successful sequencing was realized in 46/47 libraries (97.9%), with a 12,272–294,265 range of total reads (median value 83,723). *KRAS* p.Gly13Asp and *TP53* p.Arg280Lys variants were successfully found in the MDA triplicates, as well as *PIK3CA* Glu545Lys in MCF-7 (except for one combination), regardless of the number of tumor cells present in a pool (Table 2).

In detail, considering sequencing data from the pools composed of only MDA/MCF-7 cells, a high concordance was achieved for the MAFs of all hotspots, compared to the expected ones, especially when libraries derived from two cells were studied (Table 2). 

When libraries from combined pools of MDA/MCF-7/leukocytes were analyzed, *KRAS*, *TP53,* and *PIK3CA* variants were still detectable in all the replicates (Table 2). These results indicate that, although mutated alleles deriving from MDA/MCF-7 cells were mixed with the wild-type alleles from leukocytes, a high level of sensitivity was maintained, as shown in the 1:4 MDA/MCF-7/leukocytes ratios. 

Robustness and sensitivity of the optimized NGS protocol were sustained by the detection of the Glu545Lys variant present in heterozygous mutational status (data not shown), as well as in the combination pools, where the mutated allele was heavily diluted within wild-type ones. 

However, at least in some combinations, a loss of library performance may be noticed, which created some unbalance between the expected and observed frequencies (Table 2)—as particularly evident in the 3:2 and 2:3 MDA/leukocytes ratios—and a loss of mutated alleles was evidenced in one of the duplicate pool 2:3 MCF-7 cell line (Table 2). A possible explanation for these findings may be related to the suboptimal pooling of tumor cells and leukocytes into the same tube, or may be due to an alteration of allele frequencies caused by mixture with normal cells.

A sequencing failure occurred only in one out of the 47 libraries prepared (2.1%), confirming again the strength of the proposed modified NGS protocol. Furthermore, the protocol displayed 100% specificity, as no other mutations were found except for those expected and all leukocytes did not show any false positive calls.

In conclusion, these data demonstrate that the changes introduced in the original OBcfRAv2 protocol did not affect its performance.

#### 2.1.3. A glance at Variant Calling and Coverage Metrics 

In order to better define the coverage metrics of *KRAS* p.Gly13Asp, *TP53* p.Arg280Lys, and *PIK3CA* Glu545Lys variants, an in-depth analysis of the Variant Caller Format (VCF) files was performed. In particular, attention was focused on the Variant Family Size Histogram (VFSH) paired values, a parameter indicating the read count (Allele Read Coverage), and on the number of DNA template molecules (i.e., the tagged original targets), where the count was observed (Allele Molecular Coverage). Overall, this parameter indicates how many reads were attributed to each original mutated DNA molecule and specifies the robustness and reliability of the variant calling. It is appropriate to specify that the VFSH only describes the mutated DNA molecules which passed the quality check of the Torrent Variant Caller (TVC, version 5.10) plugin (i.e., those having at least three reads).

When we analyzed the VFSH values for the mutated libraries (36/47), we found a median value of 11.5 reads obtained for each DNA molecule containing *KRAS* p.Gly13Asp, 18.5 reads for those with *TP53* p.Arg280Lys variants, and 16 reads for *PIK3CA* Glu545Lys (Figure 3). The lower and upper quartiles of the median value of reads were five and 19.25 for *KRAS*, eight and 22.25 for *TP53,* and 11.5 and 20 for *PIK3CA* variant alleles. 

### 2.2. Optimized Molecular Tagging NGS Workflow Revealed to be Reliable for Molecular Analysis of CTCs 

Subsequently, we tested the modified protocol in a BC clinical setting. To this purpose, blood samples of eight BC patients were collected and CTCs were isolated using the DEPArray^TM^ system. CTCs were identified as medium-sized cells (diameter > 10 μm) with a high nucleus–cytoplasm ratio (n/c > 0.5), CD45-expression-negative, and nuclei-staining-positive; fluctuation of CD44 marker expression was observed, as well as some variability in EpCAM expression (Figure 4).

As the analysis of possible CTCs subpopulations (i.e., with epithelial and/or mesenchymal characteristics) was beyond the scope of this study, CTCs recovery was mainly realized on the basis of two parameters: morphology (shape and round features) and the complete absence of CD45 expression. Thirteen pools of different numbers of CTCs (range: 2–6 cells) were recovered and used for subsequent NGS experiments. Libraries were successfully prepared and sequenced from all the 13 CTCs pools (Table 3), thus showing that even if this analysis was limited in number, the optimized molecular tagging NGS workflow was robust when applied to “real” CTCs. Quantification of libraries revealed a yield range of 11–107 pMol, with an optimal performance (in terms of total produced reads) ranging from 20,633 to 294,305.

When available, two and/or three different pools of CTCs from the same patient were analyzed. In particular, one sample was tested for Patients 5, 7, 9, and 12; two samples for Patients 18 and 51; and, finally, three independent pools were tested for Patient 10 (Table 3). Analyses performed in more than one replicate showed concordant results.

Table 3 shows the gene variants found in six out of the eight patients studied. In particular, an intronic variant in the *TP53* gene (c.1100 + 30A > T, chr17:7573897) was present in the CTCs of Patients 5, 12, and 24; while patient seven displayed a synonymous variant in the *TP53* gene (p.Arg213=). Interestingly, both the cellular pools containing three and six CTCs from Patient 51 were characterized by a *PIK3CA* variant (p. Hys1047Arg).

We further investigated whether CTCs pools and matched BC tumor tissue shared the same mutational profile (Table 3). To this end, BC tissues were analyzed in targeted NGS by using a custom panel comprised of 20 genes and, among others, all the DNA target regions covered by OBcfRAv2. When the sequencing analysis was restricted to these common regions, six out of eight cases shared mutations between CTCs and matched BC tissues. In detail, 5/8 cases (62.5%; i.e., Patients 5, 7, 9, 12, and 51) displayed a complete mutational concordance between CTCs and tissues; indeed, a partial concordance was achieved in one patient (Patient 24), as her breast tissue showed an additional *TP53* variant (p.Arg248Trp) which was not found in the two CTC pools investigated. 

The remaining two BC patients (Patients 10 and 18) were completely discordant, as the results for both showed mutation in the matched tissues but not in the CTCs pools analyzed. In both patients, tissue variants occurred in the *TP53* gene, even if in different hotspots—p.Arg248Gln and p.Cys275Leufs, respectively (Table 3). These results could be explained by the intrinsic nature of CTCs, which may reflect completely the mutational profile of the bulk tumor, but could also diverge.

Finally, the overall analysis of NGS data from BC tissues revealed additional mutations, mainly in *ERBB2* and *PIK3CA* and occurring in 6/8 BC cases, which were not otherwise demonstrable in the matched CTCs as the genomic mutated positions did not overlap with those covered by the OBcfRAv2 panel.

## 3. Discussion

The possible clinical utility of the CTCs obtained by liquid biopsy in routine laboratory practice is still under debate, mostly due to the lack of robust and standardized procedures for their detection and relative molecular characterization. However, especially in BC settings, CTCs have gained a valuable clinical role, as their presence into the bloodstream represents a negative prognostic factor in terms of malignancy and, moreover, the study of their mutational profile provides precious information about actionable and/or resistance genetic alterations [26,27,28].

To present, many of the published protocols about CTCs genomic analysis include a preamplification WGA step, prior to “omic” techniques, with the aim of increasing the few DNA starting copies isolated from a single-cell or pools of few cells [11,12,13,14,15,16]. However, WGA procedures introduce important biases (i.e., allelic dropout, chimeric molecules, low uniformity, and polymerase errors), thus making the interpretation of downstream data both difficult and ambiguous [17,18,19,20,21]. The current major efforts have been focused on the optimization of different commercial WGA protocols coupled with NGS procedures [13,14,29,30].

Recently, molecular tagging NGS technology has found its main application in the liquid biopsy field, particularly for the research and detection of exiguous ctDNA molecules [31,32]. This is possible thanks to the use of unique molecular barcodes which allow us to intercept and “tag” rare DNA alleles, thus providing an effective solution to track and distinguish sequencing reads from the original DNA molecule (Figure 5). 

In this way, the artifacts generated during library amplification are filtered during variant calling and true mutations are correctly detected. This provides a great advantage over traditional amplicon-based NGS approaches, as the latter is more error-prone, especially when low-frequency mutations are studied and more PCR cycles are needed, further increasing the difficulty of discriminating false positives.

With these premises, our study proposes a new NGS workflow based on molecular tagging technology to study the molecular profile of CTCs, with the main goal of avoiding WGA procedures and their relative drawbacks.

Firstly, in the optimization phase of our study, an MDA cell line spiked into healthy donor blood was used as a CTCs surrogate, in order to simulate real-life conditions, and sorted using a DEPArray^TM^ system. To perform the downstream molecular analysis, we used the OBcfRAv2, a molecular tagging-based NGS panel which was originally developed to search for mutations in ctDNA of BC patients. We initially tested the robustness of this assay when applied to single or pooled MDA cells by adding 5 ng of normal genomic DNA, as this step would intentionally raise the amount of starting nucleic acids to a sufficient quantity for preparation of the NGS libraries.

We obtained optimal sequencing results in each of the MDA pools analyzed, thus demonstrating that high specificity and sensitivity were maintained, even when the maximum dilution of mutated DNA molecules was reached (one cell).

In our opinion, this procedure works well in the case where the mutations followed are known; however, we expect that, in absence of mutations, the distinction between wild-type DNA molecules derived from normal genomic DNA and those from tumor cells is not allowed, thus precluding the translational value of the proposed protocol. 

Consequently, a new procedure was developed in order to perform WGA-free library construction, which does not include the addition of any external DNA (other than that derived from cell lysates). The procedure was tested in spike-in experiments using both MDA and MCF-7 cell lines, harboring the different mutations included in the chosen NGS panel.

In this case, massive sequencing data demonstrated that, although the decrease in the starting DNA amount was important (i.e., nanograms versus picograms), the technology was still able to detect *KRAS* p.Gly13Asp, *TP53* p.Arg280Lys, and *PIK3CA* Glu545Lys variants in all the libraries obtained from cellular preparations of two, four, and five MDA/MCF-7 cells (Table 2).

Moreover, the chance to confirm the strength of the optimized NGS workflow was offered by sequencing analysis of MDA/MCF-7/leukocytes combined pools. In fact, although mutated DNA alleles (derived from MDA/MCF-7 cells) were further diluted by wild-type DNA alleles (from leukocytes), the presence of *KRAS* p.Gly13Asp, *TP53* p.Arg280Lys, and *PIK3CA* Glu545Lys variants were still detected in all cellular combinations (even in the 1:4 MDA/MCF-7/leukocytes pools). 

We have previously confirmed the presence of these mutations in related cell lines by Sanger sequencing (data not shown) and found that both *KRAS* p.Gly13Asp and *TP53* p.Arg280Lys were homozygous in MDA, whereas *PIK3CA* Glu545Lys was heterozygous in MCF-7. This represented an additional challenge to overcome for the whole procedure, including sorting, library preparation, and a bioinformatic variant calling process. In fact, in such a situation (where only one of the two alleles in a tumor cell contained the mutation), the successful detection in even the 1:4 MCF-7/leukocytes triplicate further demonstrated the robustness of the proposed protocol. 

However, some loss of library performance was noticed for the 2:3 and 3:2 combinations, as well as the underestimation of mutated alleles with respect to wild-type ones. This may be related to suboptimal pooling of MDA/MCF-7 cells and leukocytes in the same pool, or the (intentional) contamination from normal cells may have caused unpredictable MAFs.

Other evidence supported the robustness of the optimized workflow. First, DEPArray^TM^ cell recovery was realized in 0.2 mL tubes and the subsequent steps leading to amplification of tagged DNA molecules were conducted in the same reaction tube, thus warranting no or minimal loss of the already exiguous genetic material. Second, we detected no variants different from the expected ones in all of the MDA/MCF-7 pools tested. Although our findings were slightly different compared to the literature (especially regarding the MDA cell line [13,33,34]), the relatively small panel size used in our experiments must be taken into account, as it did not cover large DNA target regions compared to other NGS panels. Third, no false positives were found in the sequencing analysis of the libraries derived from isolated leukocytes, thus confirming that consistent data on the specificity of the NGS process were obtained (Table 2). Fourth, our results showed that the read counts per mutated DNA molecule (median of 11.5 for *KRAS* p.Gly13Asp, 18.5 reads for *TP53* p.Arg280Lys, and 16 for *PIK3CA* Glu545Lys) were consistently higher, compared to the minimum value suggested by the manufacturer to enable a valid variant calling (i.e., equivalent to at least “3” reads). 

The proposed workflow, beyond its technical aspects, also has interesting and important cost-efficiency related advantages. In fact, the reagent volume reduction allowed us to triplicate the number of reactions possible for the OBcfRAv2 kit, thus raising the number of libraries to a total of 24 preparations (compared to the eight proposed by the manufacturer). Moreover, the procedure supports “low-coverage sequencing”, as the scarce genetic material used as input does not require a high level of read depth. 

We are aware that the proposed NGS workflow needs further refinements to be definitively applied; for example, a critical point in the proposed procedure regards the allele counts per target region amplified (Total Molecular Coverage). Indeed, the observed read counts were usually higher than those expected (data not shown). We speculated that this could be due to misreading of the molecular tags during the amplification cycles, causing an event known as “barcode resampling” which resulted in an increase of the allele counts [35]. However, this event did not seem to alter the accuracy of the overall variant calling.

In the second phase of our study, the possible translational value of the proposed workflow was proved in a “real” clinical setting. This approach could have great value in clinical settings, where liquid biopsies are now starting to be included in the validated tests panel used to detect mutations for target therapies, such as the use of alpelisib for the treatment of patients with hormone receptor-positive, HER2-negative, and *PIK3CA*-mutated advanced or metastatic BC progressed after previous endocrine therapy as monotherapy [36]. In this context, one of the most intriguing aspects of our protocol is the possibility to study CTCs molecular profiles in women who are progressing many years after the primary tumor and who do not have biopsy from relapsed mammary tissue or from other progression sites. In these cases, we believe that this molecular analysis will allow us to discover emerging mutations, which is important for designing treatment approaches. In this view, comparative studies on ctDNA and CTCs will shed new light on the importance of studying both compartments in the BC setting.

To test the protocol’s applicability in a real clinical setting, it was applied to 13 different pools of CTCs isolated from eight BC patients. We found that 7/13 BC CTCs samples had a mutation occurring; mainly in the *TP53* and *PIK3CA* genes, as these genes are the most frequently mutated in BC [37,38,39]. 

Comparison between CTCs and matched primary BC tumor tissues demonstrated complete concordant mutational data in 5/8 BC patients; whereas two patients were discordant, and a partial match was found in one patient. These data were not surprising and are in line with the current literature, showing a certain rate of heterogeneity in BC [30,40,41,42]. 

It is necessary to note that the NGS panel utilized for neoplastic tissues sequencing covers a larger number of target regions, compared to the OBcfRAv2 kit; thus, some tissues displayed other mutations not identified in the corresponding CTCs. 

To the best of our knowledge, this is the first work where a molecular tagging-based NGS without a preamplification step was applied for the molecular analysis of single CTCs, even if some other studies have adapted amplicon-based NGS technology to the mutational profiling of CTCs [23,24]. Although the idea to skip WGA is not novel, the contributions of more laboratories are still required in order to finally obtain reliable workflows for CTCs purposes. In this regard, we believe that our approach is user-friendly, robust, and easy to implement in a routine setting by molecular clinical laboratories, providing the intriguing possibility to apply an already standardized and accessible (commercial) assay for both ctDNA and CTCs molecular analyses. 

Compared to other, highly sensitive approaches which are applied for liquid biopsy purposes, such as BEAMing (Beads, Emulsions, Amplification, and Magnetics) assay [43], this procedure allows the use of minimal input amount of genetic material (picograms) and the examination of many DNA target regions simultaneously. Instead, BEAMing is able to test only few target regions at once and, consequently, are not exactly suitable when dealing with comprehensive mutational investigations of CTCs. 

## 4. Materials and Methods 

### 4.1. Method Optimization

#### 4.1.1. Cell Line, Spiking Experiments, and DEPArray^TM^ Cell Sorting

BC-derived MDA (triple-negative) and MCF-7 (HR/PgR+, HER2−) cell lines were cultured in complete DMEM medium (Gibco^®^, Waltham, MA, USA). 

Among the variants reported to occur in MDA and MCF-7 cells (see ATCC website, https://www.lgcstandards-atcc.org/~/media/03E3D0D9F0C74FE593CE7D010F982CA2.ashx), two specific variants were investigated for MDA (i.e., the c.38G > A, p.Gly13Asp in exon 2 of the *KRAS* gene and the c.839G > A, p.Arg280Lys in exon 8 of *TP53*) and one for MCF-7 (the c.1633G > A, p.Glu545Lys in exon 10 of the *PIK3CA* gene), as their hotspot positions were also covered by the NGS panel utilized for sequencing analyses below.

A total of 1–10 × 10^3^ MDA and MCF-7 cells were separately spiked in 7.5 mL healthy donor blood and enrichment was performed using immunologic negative selection (adapted from Bulfoni et al. [44]). Briefly, after red blood cell lysis (Red Blood Cell Lysis Solution, Miltenyi Biotec, Bologna, Italy), cells were incubated with CD45 and CD235 A (Glycophorin A) MicroBeads antibodies (Abs, Miltenyi, Bologna, Italy) and depletion of leukocytes and red blood cells were realized through magnetic LD separation columns. The negative flow-throughs containing MDA/MCF-7 cells were then labeled with anti-EpCAM (clone 9C4; BioLegend, Koblenz, Germany) fluorescein isothiocyanate- (FITC), anti-CD44 phycoerythrin (PE, Clone 515 Becton Dickinson, Milano, Italy), anti-CD45 allophycocyanin (APC) conjugated (Clone 5B1 Miltenyi, Bologna, Italy), and Hoechst 33,342 (Merck Life Science, Milano, Italy). Then, cells were fixed and permeabilized (Inside stain kit, Miltenyi, Bologna, Italy) for complete nuclei staining, reduced to a volume of 13–14 μL after washing with SB115, loaded into an A300K V2 DEPArray^TM^ cartridge and, finally, selected and sorted on a DEPArray^TM^ system (Menarini Silicon Biosystems, Bologna, Italy). 

MDA/MCF-7 cells were identified on the basis of morphological (as medium- or large-sized cells) and phenotypic features (such as CD44/EpCAM positivity and CD45 negativity), and isolated individually or in pools composed of 2, 3, 4, or 5 cells. Combined pools containing MDA or MCF-7 and leukocytes were arranged according to different cells: leukocytes ratios of 4:1, 3:2, 2:3, and 1:4 were used. Leukocytes (bright CD45 positive small size cells) were also recovered grouped in two and five cells.

Cells were collected into 0.2 mL tubes and stored in a final volume of 1–2 μL of PBS at −20 °C until molecular analysis.

#### 4.1.2. Evaluation of DNA Quality

DNA from single or pooled cellular preparations of both MDA cells and leukocytes were assessed by a WGA assay (Ampli1 WGA kit, Silicon Biosystems, Bologna, Italy) and multiplex PCR with the Ampli1 QC kit (Silicon Biosystems, Bologna, Italy), which amplified four different chromosomal regions with different lengths (91, 108–166, 299, and 614 bp, respectively with 12 p, 5 p, 17 p, and 6 p chromosomes), as previously described [25]. The number of bands obtained by each PCR product was visualized on a 3% agarose gel through UV light.

#### 4.1.3. Canonical NGS Workflow Assessment 

DNA from single or pools of MDA cells was obtained by treatment with 2 µL of Lysis Reaction Mix (Ampli1 WGA kit, Silicon Biosystems, Bologna, Italy).

MDA cell lysates were processed together with 5 ng of normal genomic DNA from healthy donor blood and MDA/normal DNA mixtures were used for library construction with the OBcfRAv2 (ThermoFisher Scientific, Carlsbad, CA, USA), following the manufacturer’s instructions (https://assets.thermofisher.com/TFS-Assets/LSG/manuals/MAN0017065_Lung_Breast_cfTNA_Assay_UG.pdf. The assay targeted more than 150 hotspot positions relatively to 10 genes: *AKT1, EGFR, ERBB2, ERBB3, ESR1, FBXW7, KRAS, PIK3CA, SF3B1,* and *TP53* (for further information, see https://www.thermofisher.com/order/catalog/product/A35865). 

Manually prepared libraries were quantified by qPCR assay with an Ion Library TaqMan^TM^ Quantitation kit (ThermoFisher Scientific, Carlsbad, CA, USA), diluted to 50–60 pMol and then multiplexed up to 5 samples for automated template preparation on an Ion Chef^TM^ System and loaded on an Ion 530^TM^ Chip (ThermoFisher Scientific, Carlsbad, CA, USA). Sequencing runs were performed on an Ion Torrent GeneStudio^TM^ S5 (ThermoFisher Scientific, Carlsbad, CA, USA), according to the manufacturer’s user manual.

#### 4.1.4. Optimized Molecular Tagging NGS Workflow

Libraries were prepared using directly the 3 µL cell lysates from isolated cells as a source of DNA. The entire protocol for library preparation with OBcfRAv2 was modified to adjust the amplification of small quantities of DNA input into a small reaction volume. 

All reactions used for library construction were set up with a 1:3 dilution reduced volume (see details in Appendix A). Thermal conditions and number of amplification cycles were maintained as the canonical original protocol suggested (Appendix A). 

Manually prepared libraries were quantified, diluted, and used for template preparation, as described in Section 4.1.3. The only relevant modification consisted of multiplexing up to 24 amplified libraries on an Ion 520™ Chip (low-coverage sequencing). Run sequencing was performed on an Ion Torrent GeneStudio^TM^ S5, as described above.

The overall study design and the main changes throughout the entire procedure are summarized in Table 1.

#### 4.1.5. NGS Analysis

Molecular data analysis was performed on the Torrent Suite (version 5.10) software. The Sequencing Binary Alignment Map (BAM) file of each library was generated through an automatic pipeline already set by ThermoFisher, which provides read alignment and mapping on reference hg19, thanks to the Torrent Mapping Alignment Program (TMAP). The TVC plugin was run, in order to get the VCF files and list all the mutations that passed the quality filters. Then, VCFs were visualized and interpreted using the Integrative Genomic Viewer (IGV, Broad Institute, Cambridge, MA, USA).

### 4.2. CTCs Testing

#### 4.2.1. BC Patients 

Eight BC patients were randomly chosen from those enrolled in a translational trial conducted at IRCCS Ospedale Policlinico San Martino (Citrucel Protocol approved by Regione Liguria Ethics Committee, Decision No. 0519, 4 May 2016). In particular, six patients were treated in the neoadjuvant settings, one patient in the adjuvant setting, and one patient in the metastatic BC stage. All patients gave their written informed consent before their enrolment. 

#### 4.2.2. Enrichment, Fixation, and Sorting of CTCs from BC Patients

Blood draws were collected, in each case, before systemic treatment. BC CTCs were isolated according to the same enrichment, fixation, and sorting procedures used for MDA/MCF-7 cell lines (see Section 4.1.1). Due to the variability of EpCAM and/or CD44 expression in patient cells, DEPArray^TM^ selection of putative CTCs was mainly based on morphology (oval shape and round features), positive Hoechst staining, and the complete absence of CD45 expression; as well as cell diameter higher than 10 μm and a nucleus–cytoplasm ratio > 0.5.

Cell sorting was differentially arranged in various pools through the 8 patients, depending on the number of CTCs identified in their blood (pool range: 2–6 cells). Recovered pools were stored at −20 °C until molecular analysis.

NGS tests on CTCs were performed using the optimized molecular tagging NGS workflow described in Section 4.1.4 and the analysis was conducted as in Section 4.1.5.

### 4.3. NGS of BC Tissues

A custom panel based on the Ion AmpliSeq^TM^ technology was designed to study common alterations in BC, with overlapping target regions present in the OBcfRAv2. Specifically, it is comprehensive of 291 amplicons distributed within 20 genes (i.e., *TP53*, *PIK3CA*, *ERBB2*, *ERBB3*, *ERBB4*, *ESR1*, *MCL1*, *GATA3*, *PTEN*, *CCND1*, *KRAS*, *AKT1*, *CDH1*, *MAP2K4*, *SF3B1*, *FBXW7*, *MAP3K1, PIK3R1, EGFR,* and *FGFR1*). 

Matched Formalin-Fixed Paraffin-Embedded (FFPE) tumor tissue sections (5 μm thickness) were used for automated genomic DNA extraction with a QIAsymphony instrument (Qiagen, Milano, Italy). Tumor tissue enrichment was performed by macro-dissection of areas containing at least 50% of neoplastic cells.

A total of 10–15 nanograms of DNA were quantified by a Qubit^TM^ 3.0 fluorometer, used for automated library construction on the Ion Chef^TM^ with an Ion AmpliSeq^TM^ Kit for Chef DL8 (ThermoFisher Scientific, Carlsbad, CA, USA).

Fifty pMol of eight mixed libraries were used for automated template preparation on the Ion Chef^TM^ System and loading on an Ion 520^TM^ Chip. The run sequencing step was performed on the Ion Torrent GeneStudio^TM^ S5.

NGS data analysis was executed in the Ion Torrent Suite and the Coverage Analysis and TVC plugins were run. VCFs were visualized on IGV and variants annotation was performed with the Ion Reporter Software (version 5.10, ThermoFisher Scientific, Carlsbad, CA, USA).

## 5. Conclusions

This study has demonstrated, for the first time, the potential application of a molecular tagging NGS panel, which is already standardized for other liquid biopsy purposes, as a feasible mutational approach for CTCs sequencing. Here, we demonstrated that accurate tumor cells isolation by the DEPArray^TM^ System coupled with a precise and sensitive NGS procedure can represent not only an alternative strategy for CTCs mutational profiling, but also a reliable WGA-free method.

In this direction, these findings open the way to future translational applications of the proposed workflow in precision medicine laboratories which deal with the liquid biopsy field, as well as further supporting the importance of CTCs as a useful tool for prognostic, diagnostic, and therapeutic applications.

## Figures and Tables

**Figure 1 ijms-21-04364-f001:**
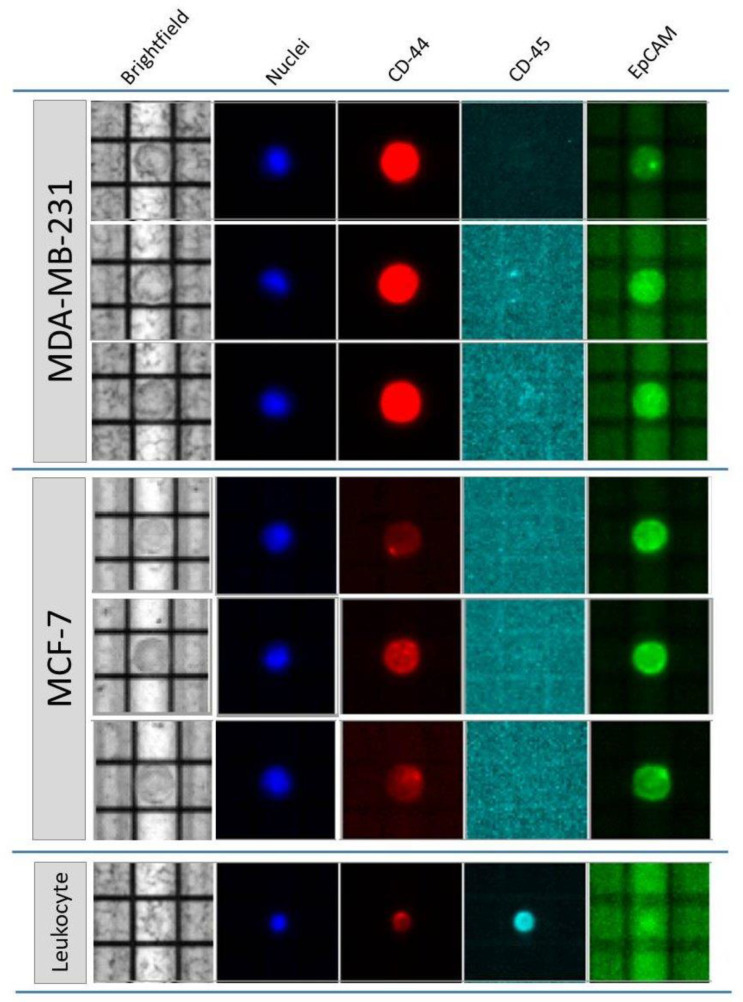
Analysis of MDA/MCF-7 and leukocyte cells on DEPArray^TM^. The panel shows the morphological and fluorescent characteristics of MDA-MB-231 (MDA) or MCF7 cells spiked into healthy donor blood and isolated on DEPArray^TM^. MDA cell is typically CD45-negative and Hoechst-positive (light blue and blue stains). In the first lane, a low EpCAM-positive MDA cell can be observed. MCF7 cells are all EpCAM-positive and show different expressions of CD44, almost negative (lines 4 and 6), and occasionally positive (line 5). In comparison, below, a CD44-positive/CD45-positive and Hoechst-positive leukocyte is shown.

**Figure 2 ijms-21-04364-f002:**
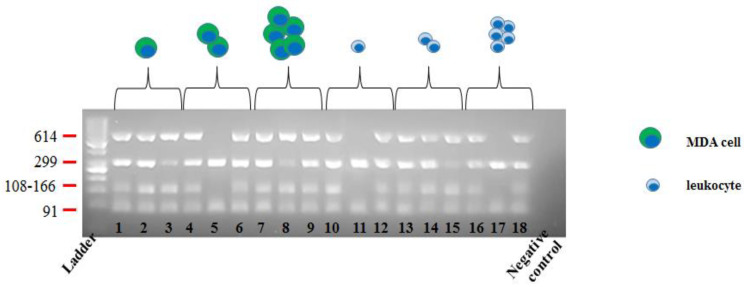
DNA integrity testing in sorted MDA/leukocytes cells. Multiplex PCR was performed and four PCR products of different lengths from four chromosomal regions were amplified. The number of bands obtained by each PCR product was visualized on a 3% agarose gel through ultraviolet light. Each lane represents a single/cellular pool.

**Figure 3 ijms-21-04364-f003:**
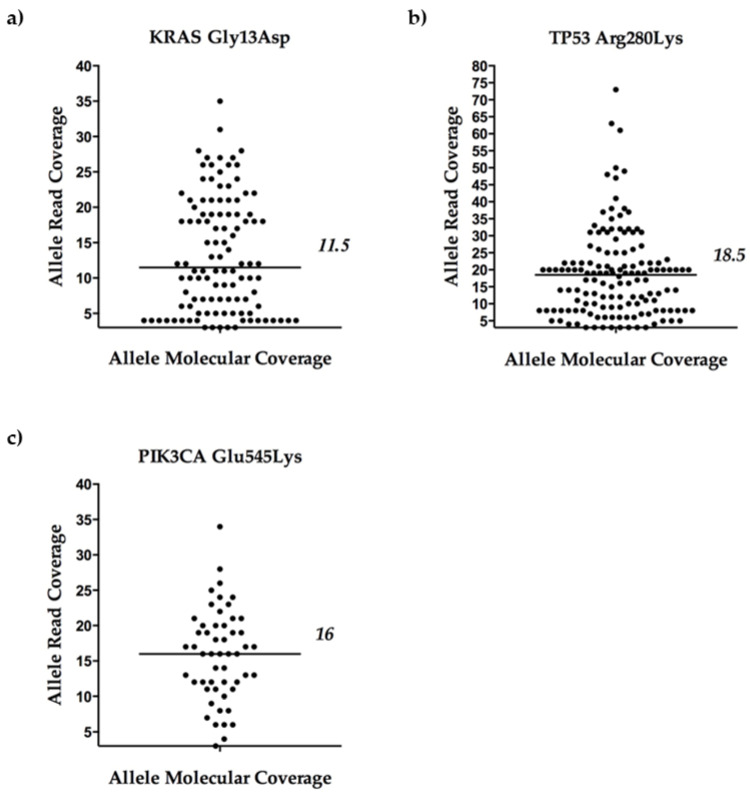
Coverage metrics of *KRAS* Gly13Asp, *TP53* Arg280Lys, and *PIK3CA* Glu545Lys variants. Scatter dot plots in (**a**,**b**) show the Allele Read Coverage and the Allele Molecular Coverage for *KRAS* Gly13Asp and *TP53* Arg280Lys, respectively, as found in MDA pools, and in (**c**) for *PIK3CA* Glu545Lys in MCF-7. Each dot represents a single molecular DNA molecule on the *x*-axis (i.e., Allele Molecular Coverage) and the corresponding number of reads on the *y*-axis (i.e., Allele Read Coverage). The origin of the axes was set to 3, corresponding to the default minimum value of reads to enable a variant calling.

**Figure 4 ijms-21-04364-f004:**
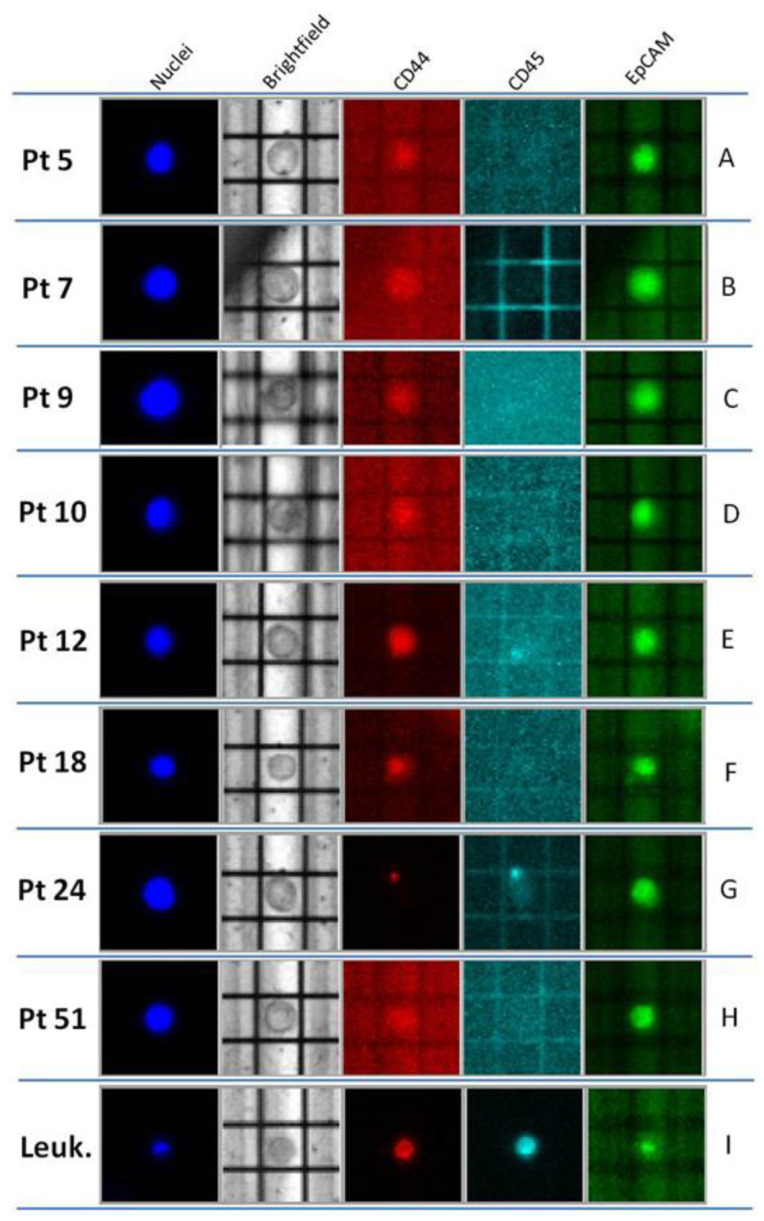
Representative Circulating Tumor Cells (CTCs) recovered from breast cancer (BC) patients on DEPArray^TM^. CTCs (**A**–**H**) are identified as round, middle-sized cells, which are Hoechst and EpCAM-positive and CD45 negative. A low CD44 expression is appreciated only in Pt. 12. In (**G**), the CD45 fluorescent dot may represent a non-nuclear component of blood (i.e., erythrocyte or platelet). Leukocyte in (**I**) is CD44- and CD45-positive.

**Figure 5 ijms-21-04364-f005:**
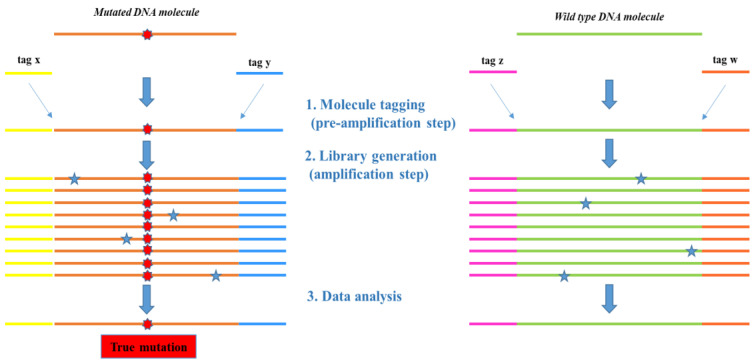
Overall principle of sequencing based on molecular tagging NGS approach. Unique molecular tags bound to gene-specific primers are added into every DNA molecule (Step 1) prior to the library amplification step, in order to keep track of possible PCR and/or sequencing artifacts (blue stars). All of the reads (Step 2) are then collapsed with the same genomic coordinates and molecular tag into a single representative read (Step 3), corresponding to an original mutated (on the left) or wild-type (on the right) DNA template.

**Table 1 ijms-21-04364-t001:** Comparison between canonical and optimized molecular tagging Next-Generation Sequencing (NGS) workflows.

	Canonical Molecular Tagging NGS Workflow			Optimized Molecular Tagging NGS Workflow								
Types of starting isolated cells	MDA cells (single or pools)			MDA ^a^/MCF-7 ^b^ cells (only pools)			MDA/MCF-7: leukocytes (combined pools)				Leukocytes (only pools)	
n of cells	1	3	5	2	4	5	4:1	3:2	2:3	1:4	2	5
5ng of DNA from healthy donor leukocytes	+	+	+	−	−	−	−	−	−	−	−	−
OBcfRAv2 volume reagents reaction	As recommended by manufacturer			Modified: 3× reduction of volume reagents								
Thermal PCR conditions	As recommended by manufacturer			As recommended by manufacturer								
Libraries quantification	qPCR			qPCR								
Libraries multiplexing on chip	Five libraries on Ion 530^TM^ Chip			24 libraries on Ion 520^TM^ Chip (low-coverage sequencing)								

^a^ MDA sorting experiments were all performed in triplicate. ^b^ MCF-7 sorting experiments were performed in duplicate, except for pools comprising two tumor cells and 1:4 MCF-7/leukocytes combined pools, which were performed in triplicate. + addition of 5 ng of normal genomic DNA. − no addition of normal genomic DNA.

**Table 2 ijms-21-04364-t002:** Variant calling results from MDA, MCF-7, and leukocytes libraries.

		MDA-MB-231						MCF-7		
		*KRAS* Gly13Asp			*TP53* Arg280Lys			*PIK3CA* Glu545Lys		
Pool Type	*n* of Cells	Expected MAFs (%)	Mean Observed MAFs (%)	Median Allele Molecular Coverage	Expected MAFs (%)	Mean Observed MAFs (%)	Median Allele Molecular Coverage	Expected MAFs (%)	Mean Observed MAFs (%)	Median Allele Molecular Coverage
**tumor cells**	2	100	100	7	100	100	10	50	38	2
	4	100	100	8	100	100	12	50	66.7	2.5
	5	100	100	10	100	95.8	14	50	43.7	6.5
**tumor cells: leukocytes**	4:1	80	76.2	5	80	87.5	7	40	33.3	6
	3:2	60	83.4 ^a^	2 ^a^	60	57.5 ^a^	6 ^a^	30	50	3
	2:3	40	37.2	5	40	57.6	5	20	36.4 ^b^	4 ^b^
	1:4	20	24.4	2	20	45.1	3	10	14.8	1
**Leukocytes**	2	0	0	0	0	0	0	0	0	0
	5	0	0	0	0	0	0	0	0	0

^a^ Calculated on 2 replicates, due to a library sequencing failure. ^b^ calculated on a single replicate.

**Table 3 ijms-21-04364-t003:** Mutational comparison between CTCs and corresponding tumor breast tissue.

Pt	CTCs	Concordance CTC/Tissue	BC Tissue Mutations
(Number Cells/Reaction)	Mutations (OBcfRAv2)	Detectable by OBcfRAv2 and Custom Panel	Detectable Only by Custom Panel
*5*	3	*TP53* c.1100 + 30A > T	yes	*TP53* c.1100 + 30A > T	*ERBB2 p.Ile654Val*ERBB2 p.Ile655Val*ERBB2 p.Pro1170Ala*
*7*	3	*TP53* p.Arg213=	yes	*TP53* p.Arg213=	*PIK3CA* p.Asn345Lys *ERBB2* p.Ile655Val
*9*	3	not found	yes	not found	*PIK3R1 p.Met326Ile* *ERBB2 p.Pro1170Ala*
*10*	2	not found	no	*TP53* p.Arg248Gln	
2	not found
4	not found
*12*	5	*TP53* c.1100 + 30A > T	yes	*TP53* c.1100 + 30A > T	*ERBB2* p.Pro1170Ala
*18*	3	not found	no	*TP53* p.Cys275Leufs	*ERBB2* p.Pro1170Ala
3	not found
*24*	2	*TP53* c.1100 + 30A > T	partial	*TP53* p.Arg248Trp*TP53* c.1100 + 30A > T	*ERBB2* p.Pro1170Ala
5	*TP53* c.1100 + 30A > T
*51*	3	*PIK3CA* p. Hys1047Arg	yes	*PIK3CA* p. Hys1047Arg
6	*PIK3CA* p. Hys1047Arg

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
