# Peer review of "Optimization of a WGA-Free Molecular Tagging-Based NGS Protocol for CTCs Mutational Profiling"

_ijms, 2020, doi:10.3390/ijms21124364_

Round 1
Reviewer 1 Report
The authors present a study of WGS of DEP array isolated CTC without the need for whole genome amplification. The technique was first validated with cell line spike in experiments using the MDA-MB-231 cell line and tested in blood samples from breast cancer patients.
The cell line used was not the most appropriate as it only harbors mutations in 2 of the genes included in the sequencing panel. MDA-MB-231 has over 300 somatic mutations, although only the KRAS and Tp53 genes overlapped with somatic mutation profile of the cell line and the panel used for sequencing. It would have been better to use additional cell lines or innfact, a different cell line with mutations in other genes included in the panel such as BRAF and AKT.
These types of studies are more useful to identify low frequency mutations that drive metastasis and could be therapeutic targets. However, the authors have only managed to identify the most frequent mutations in these tumors and the corresponding CTC.
It is difficult to determine the clinical utility of this method over primary tissue based studies or sequencing or PCR based approaches to mutation detection in isolated CTC such as BEAMing. In fact, the kit used for sequencing can also be used for cfDNA and this study did not clearly show how their CTC strategy is superior to the analysis of cfDNA. Where the authors able to identify CNVs in CTC and spike in experiments that can theoretically be performed with the sequencing kit used?
The article is poorly presented and difficult to follow and needs to be reviewed by a scientific writer. The authors need to clearly define why they have performed this study and also show how this technique is superior to what is currently available and explain the potential clinical utility. The study is potentially interesting but needs extensive changes to be considered for publication.
Author Response
Thanks to the Reviewer for the comments and suggestions above.
The main purpose of the paper was to demonstrate the technical feasibility of a Next Generation Sequencing (NGS) optimized protocol based on molecular tagging technology applied to CTCs mutational investigation. To this end, breast cancer (BC) was used as target model to validate the proposed NGS protocol since many studies have already demonstrated the importance of CTCs search and count in this clinical setting. Consequently, the NGS panel type utilized in the validation was the one best fitting our model since created in order to search specific somatic mutations in circulating tumor DNA from BC patients. The panel targets more than 150 hotspots genomic positions belonging to 10 genes, i.e. AKT1, EGFR, ERBB2, ERBB3, ESR1, FBXW7, KRAS, PIK3CA, SF3B1 and TP53, even if we were aware that the used panel doesn’t cover the overall mutational landscape of BC.
Following our target model, we then selected the MDA-MB-231 (MDA) cell line for the procedure setup/optimization, as it represents an accepted in vitro surrogate for triple negative breast cancer subtype. We agree that the MDA could not be the most appropriate cell line for such a validation study, since only two among other described mutations (KRAS Gly13Asp and TP53 Arg280Lys) are included in the used panel. Therefore, to overcome this issue and meet the Reviewer’s request, we performed an additional spiking in experiment and DEPArray sorting using another mammary derived cell line, the MCF-7, that represents the most used model of hormone positive/HER2 negative BC. As already described, this cell line is characterized, among all, by a mutation in exon 10 of PIK3CA, i.e. Glu545Lys, that is targeted by the NGS panel used in our study. Thanks to Reviewer suggestion, this additional test allowed us to confirm how robust and specific is the modified NGS protocol we proposed. Indeed, in contrast to KRAS Gly13Asp and TP53 Arg280Lys which were in a homozygous status in MDA (Sanger sequencing, not shown), the PIK3CA Glu545Lys harbored by MCF-7 was in a heterozygous status (Sanger sequencing, not shown) and consequently its detection needed more specificity and sensitivity levels (see also Discussion, lines 330-336). We were able to find this mutation in 15/16 pools of the sorted MCF-7 cells and most of all even in the MCF-7:leukocytes combinations (see Results section, Table 2 line 170, and text, lines 183-185). No other additional mutations were found, thus showing again that methodology is far accurate.
It is true that the used panel doesn’t provide a genome wide analysis of the overall alterations possibly occurring during the BC disease progression and harbored by CTCs, but it is equally true that this preliminary paper had the goal to validate the feasibility of the proposed modified NGS protocol from technical and analytical points.
We agree with the reviewer that other technologies are currently used for liquid biopsy. However, we believe that the proposed workflow “allows the use of a minimal input amount of genetic material (picograms) and the examination of many DNA target regions simultaneously. Instead, BEAMing is able to test only few target regions at once and, consequently, are not exactly suitable when dealing with comprehensive mutational investigations of CTCs” (discussed in lines 309-404).
The optimized protocol we proposed is also superior in terms of costs, thus in the same reaction it is possible to test more than 150 hotspot genomic positions and again, thanks to our adjustments, it is possible to employ a kit for 8 reactions (1700 euros) up to 24 ones, with a three times cost reduction per sample (70.83 versus 212.5 euros/sample).
We apologize with the Reviewer and agree with comment “In fact, the kit used for sequencing can also be used for cfDNA and this study did not clearly show how their CTC strategy is superior to the analysis of cfDNA “. Now, in lines 376-379 we tried to rewrite the sentences in order to point out and emphasize that this protocol has the great advantage, if needed, to study in parallel ctDNA and CTCs, by applying the same methodology/panel to both these “complementary” liquid biopsy biomarkers.
The criticism raised by the Reviewer about the analysis of CNV, is very important and we thank for the opportunity to discuss about. As mentioned in the discussion (lines 361-367), our system is not perfect, as it arises from the adaptation of a protocol created for other purposes, and in particular it does not allow to reach high levels of accuracy for CNVs analysis. To overcome this problem, bioinformatics-type modifications of the parameters related to the quality control filters are most likely needed. With this in mind, it is possible that this proposed collateral application of this assay for CTCs molecular characterization will interest the manufacturers to face possible modifications to let CNV calls available in this context.
Finally, we tried to better explain in the Discussion section, lines 369-379, the possible clinical utilities about the proposed approach in a clinical setting.
In summary, we think that this NGS protocol may represent an accurate alternative for CTCs mutational analysis and can be easily implemented in clinical practice by precision medicine laboratories.
Extensive english editing has been done through the MDPI English Editing Office (certificate enclosed).

Reviewer 2 Report
Giuseppa De Luca et al. have well presented the manuscript entitled “Optimization of a WGA-free molecular tagging based NGS protocol for CTCs mutational profiling”. However, the authors need to address few points to improve their manuscript. The title of reference 23 in this manuscript is “Next generation Sequencing (NGS) Analysis on Single Circulating Tumor Cells (CTCs) with No Need of Whole-genome Amplification (WGA)”. However, in the abstract (line 3), the authors mentioned that they proposed WGA-free workflow for CTCs. I am confused, what is the difference between these two papers. If I understand correctly, the authors of this manuscript have optimized the protocol but not proposed or introduced WGA-free approach – if this is true, the authors need to describe/emphasize about the parameters they have optimized for NGS protocol – the authors need to highlight these optimization details in the abstract. Or, if the authors proposed WGA-free protocol then they have to emphasize in the abstract about the novelty of this manuscript when compared with the reference 23. In summary, the authors need to rewrite the abstract by focusing on (i) the NGS protocol optimization and (ii) the improved results (or the advantages) observed with their optimized protocol when compared with the traditional (or canonical) NGS protocol.
Author Response
Thanks to the Reviewer for the criticism.
We agree that the abstract is ineffective in putting in evidence the novelty of the optimized NGS protocol used in this study. Thus, we have changed and improved the abstract as follows:
“Molecular characterization of Circulating Tumor Cells (CTCs) is still challenging, despite attempts to minimize the drawbacks of Whole Genome Amplification (WGA). In this paper, we propose a Next Generation Sequencing (NGS) optimized protocol based on molecular tagging technology, in order to detect CTCs mutations while skipping the WGA step. MDA-MB-231 and MCF-7 cell lines, as well as leukocytes, were sorted into pools (2–5 cells) using a DEPArray™ system and were employed to set up the overall NGS procedure. A substantial reduction of reagent volume for preparation of libraries was performed, in order to fit the limited DNA templates directly derived from cell lysates. Known variants in TP53, KRAS, and PIK3CA genes were detected in almost all the cell line pools (35/37 pools, 94.6%). No additional alterations, other than those which were expected, were found in all tested pools and no mutations were detected in leukocytes.
The translational value of the optimized NGS workflow is confirmed by sequencing CTCs pools isolated from eight breast cancer patients and through the successful detection of variants.
In conclusion, this study shows that the proposed NGS molecular tagging approach is technically feasible and, compared to traditional NGS approaches, has the advantage of filtering out the artifacts generated during library amplification, allowing for the reliable detection of mutations and, thus, making it highly promising for clinical use”.
Then, a more detailed description of molecular tagging technology is reported in Figure 7 and Discussion section (lines 284-299). The proof that the this methodology is highly suitable for CTCs mutational profiling is demonstrated by the fact that we never found accidental mutations neither in MDA or MCF-7 cell lines and no variants were detectable in leukocytes (used as negative controls) (Table 2, lines 170-174).
We believe that this strategy provides more reliable molecular data compared to classical NGS methods (discussed in lines 295-299) and at the same time presents more advantages compared the other currently used sensitive approaches for liquid biopsy (lines 399-404).
Extensive english editing has been done through the MDPI English Editing Office (certificate enclosed).

Round 2
Reviewer 1 Report
The authors have presented a much-improved version of the manuscript and it is clear that they have seriously considered my previous comments. The purpose of the study is now clear, an optimized NGS to detect mutations in CTC without the need for a WGA, as stated in the abstract. Furthermore, the authors have added additional data to the manuscript (PIK3CA mutation data) that has added scientific value to their study. Furthermore, in the discussion they talk about how this technology compares to the currently used and available technologies and how their system can compete with or complement these techniques. Their system has the advantage to filter our sequencing artifacts, which is important in studies with a low input. Unfortunately, there still seems to be some minor errors in the English, even though a scientific writer has proofread the article.